# Quality of Life among Couples with a Fertility Related Diagnosis

**Panagiota Dourou** [1,*] **, Kleanthi Gourounti** [1] **, Aikaterini Lykeridou** [1] **, Konstantina Gaitanou** [2] **, Nikolaos Petrogiannis** [2] **and Antigoni Sarantaki** [1]

1 Department of Midwifery, Faculty of Health and Care Sciences, University of West Attica, 12243 Athens, Greece
2 ART Unit, Athens Naval Hospital, 11521 Athens, Greece
* Correspondence: pdourou@uniwa.gr

**Abstract:** Fertility-related stress can negatively impact infertile couples' quality of life (QoL). Most previous studies have concentrated on the effects of stress and infertility on individual persons, especially women, though infertility affects the QoL of both spouses. Our research aimed to investigate the roles of infertility and stress in couples' quality of life as a single unit. The research sample consisted of 202 spouses, i.e., 101 couples, with a mean age of 39.5 years (SD = 4.9 years) undergoing fertility treatment at Athens Naval Hospital-Assisted Reproduction Unit. Data collection was completed via self-administered questionnaires: the FertiQoL International Questionnaire for measuring the quality of life in infertility and The Demographic Information and Medical History Questionnaire. Data collection was conducted between January and November 2022. Quantitative variables are expressed as mean values (standard deviation) and as median interquartile range, and qualitative variables are expressed as absolute and relative frequencies. Pearson's (r) and Spearman's (rho) correlations coefficients were used to explore the association of two continuous variables. Multiple linear regression analysis was used with dependence on the Ferti-QoL's subscales. The regression equation included terms for participants' demographics and information from their medical history. Adjusted regression coefficients (β) with standard errors (SE) were computed from the results of the linear regression analyses. All reported p values are two-tailed. Statistical significance was set to $p < 0.05$, and analyses were conducted using SPSS statistical software (version 22.0). We found that greater anxiety and depression were significantly associated with worse quality of life. Additionally, quality of life, according to Ferti-QoL, was significantly worse in women, participants with a high level of education, those with greater depressive symptoms, and those with greater state scores. Findings of this study highlight the need for implementing interventions of supportive care methods, counseling, stress reduction methods, and improving the fertility-related quality of life of infertile couples.

**Keywords:** quality of life; infertile couples; assisted reproduction; infertility psychosocial factors; quantitative methods; emotional; combat stress reactions; stress; FertiQoL international questionnaire

## 1. Introduction

Infertility is a worldwide health issue that affects millions of people of reproductive age. According to available data, infertility affects 48 million couples and 186 million individuals worldwide [1]. It is estimated that 16–26% of European women who are trying to get pregnant, experience infertility [2].

Infertility is a male or/and female reproductive system disease characterized by the inability to achieve pregnancy after 12 months or more of regular unprotected sexual intercourse. Primary infertility is the inability to achieve conception, and secondary infertility is a couple's inability to conceive after a successful conception in the past.

Infertility has been ranked as one of the most stressful situations a person can face, being comparable to divorce or the death of a family member, or even physical illnesses

such as cancer. It is estimated that one in six couples in Western societies faces infertility to some degree in their lives, and the World Health Organization calls for the recognition of infertility as a global public health problem [3].

Infertility may occur due to male or female factors, a combination of male and female factors, or unexplained [1].

Depending on the problem faced by each couple, the treatment that best suits their needs will be preferred. It can range from simple monitoring or induction of ovulation, intrauterine insemination, in vitro fertilization in a natural cycle, and micro insemination (intra-cytoplasmic sperm injection—ICSI) to selection of a sperm or egg donor, embryo transfer of a frozen fertilized egg, and even surrogacy [4].

However, environmental and lifestyle factors such as smoking, excessive alcohol consumption, obesity, and exposure to environmental pollutants have been linked to lower fertility rates [1].

Undoubtedly, clinical efforts and technology have improved therapeutic results in the treatment of infertility. However, the nature of the disease and the painstaking effort to treat it leads to the assumption that the treatment of infertility is equally important to the quality of life of infertile patients.

It is argued that for many people, infertility is a major life crisis and can cause depression, anxiety, social isolation, and sexual dysfunction [5].

Psychosocial studies among couples suffering from infertility demonstrate a high frequency of negative reactions and low satisfaction with life because of infertility and its treatment [6].

The overall well-being of the couple, the success of the treatment, the willingness to continue treatment, and the assessment of the satisfaction or lack of satisfaction that people can expect due to the success or failure of the treatment are all affected by the psychological burden experienced by the couple. Therefore, the need to measure and consider the quality of life in infertility is essential, and the use of reliable measurements could lead to better management of patients [7].

Quality of life (QoL) according to World Health Organization, is defined as a wide-ranging concept that is influenced in a complex way by physical health, mental state, personal beliefs, the level of independence of the individual, and his relationship with the environment and the conditions (social, economic, cultural, security, etc.) that it offers [8].

Thus, all the problems related to the couple's physical and mental health adversely affect their quality of life.

In addition, there is a new definition in the literature for the fertility quality of life (FertiQoL), which specifically indicates the impacts of fertility problems on various life dimensions [7]. Infertile women frequently report severe stress and poorer marital adjustment and QoL than women who conceive naturally [8]. Moreover, men may experience less intercourse satisfaction, perhaps because of the psychological pressure of trying to conceive or the forced timing of intercourse around the woman's ovulatory cycle [8].

To separate the quality of life, in its general sense, from the quality of life that concerns the requirements of medicine and clinical applications, the term health-related quality of life (HRQL) is used [9].

Therefore, the question that arises is whether psychological stress and its consequences adversely affect fertility. The research evidence regarding the statistically significant relationship between stress and infertility is contradictory [10]. However, all researchers have come to a common conclusion that regardless of the positive or negative impact of stress levels on infertility treatment, one of the primary goals of health professionals dealing with the issue should be to provide valid information, advice, and psychological support. Infertility and its treatment are a couple's problem, not an individual one, and should be treated as such. Not all individuals are equally susceptible to having their mental health disrupted by chronic infertility. However, if it is perceived that their mental balance and interpersonal relationship are disturbed, they should consult specialized health professionals [7].

There are many ways and research tools to measure the psychological burden and quality of life associated with infertility, the in vitro fertilization process, and several treatment options [10]. Tools to measure health and quality of life can be general or focus on specific diseases. They combine the subjective dimensions of the individual's personal experience and the objective assessment of external factors that can affect the individual's quality of life. The subjective indicators are related to a person's satisfaction with life, the feeling of adequacy and satisfaction from his functionality in several areas of life, satisfaction from services provided, and the possibility to participate in various activities (recreational, educational). Health, natural environment, space and living conditions, time availability, social activity, the financial ability to cover basic needs, and the security of the environment are the objective indicators that are valued [11].

Interventions to reduce and alleviate the various clinical symptoms, regardless of whether or not they contribute to an increase in pregnancy rates, are extremely beneficial to the emotional balance and harmonious relationship of the couple [12].

Data indicate that 10–15% of the global population of reproductive age experience infertility [13], yet the emotional distress of this diagnosis can be especially intense and cause an unexpected crisis [14]. The emotions experienced often lead people to isolate themselves from their partners, their families, friends, and colleagues, and to feel lonely [14].

Some studies have shown that infertility leads to intense stress and a decrease in affection between infertile couples [15]. Many people in such a situation sometimes reconsider their relationships [15]. Chachamovich et al. suggested that infertile couples experience additional stress and tension in their relationships with their spouses [16]. The results of various studies have shown that infertility is accompanied by a lack of sexual desire and marital dissatisfaction, stress, and depression in infertile couples [17].

According to research carried out on 1000 infertile couples, the mental pressures caused by infertility in both men and women are always accompanied by reduced quality of life in all aspects [18].

A clinical study showed that infertility will lead to a lower quality of life index for infertile individuals through the development of mental and social stress, reduced life satisfaction, and increased marital problems [19]. Boivin et al., in a survey conducted online in the USA, Australia, New Zealand, Canada, and the UK, developed the first global tool to measure the quality of life in men and women experiencing fertility problems (FertiQoL). In total, 1414 infertile men and women participated in this study. The FertiQoL consists of 36 sections, of which 24 sections assess the essence of treatment, 10 sections assess the quality of treatment related to the quality of life, and 2 sections assess overall life span and physical health. The results of the study showed that the FertiQoL questionnaire is a gold standard for measuring outcomes of psychological well-being in individuals and couples with unintended childlessness [20].

Therefore, research has identified various individual factors, such as gender [18,21], age [21,22], and coping strategies; factors associated with infertility, namely, type of diagnosis, type of treatment, and duration of infertility [23]; and relational factors, such as the quality of the marital relationship [24,25], influencing perceived levels of psychological health and quality of life.

Research has also highlighted that stressful life events should be considered as factors that broadly influence parameters of physical and psychological health [26,27] and perceived quality of life [28,29].

Several studies have shown that the presence of stressful events involving the family (e.g., divorce, financial problems, deaths, and maltreatment) is often reported in the biographical background of infertile couples [30,31], revealing a significant effect on relevant physical and medical parameters related to reproductive functions, such as menstrual cycle regularity [32], sperm quality [33], and pregnancy outcome after in vitro fertilization [34,35].

Many studies have been conducted on the analysis of the quality of life in infertile individuals, but most have focused on one of the spouses, especially women, and few have considered the quality of life in both spouses as a single unit. Analyzing the relation-

ships between infertile individuals and their quality of life will help us examine couples' differences in how they deal with the issue of infertility. Studies have shown that higher correspondence between the quality of life outcomes has been associated with less stress and a better ability to manage stressful situations [36].

To better understand the causes of this phenomenon, it is primarily imperative to examine closely the spouses who are undergoing fertility problems.

The purpose of this study was to analyze the relationships between infertility stress and the quality of life in a convenience sample of infertile couples and to investigate the differences between men and women in how they deal with the issue of infertility.

Some research hypotheses emerged concerning the purpose of the research and the literature review. We hypothesized that infertile couples undergoing infertility treatment would experience high levels of infertility-related stress and that infertility stress would be negatively related to infertile couples' quality of life.

The COVID-19 pandemic has claimed millions of lives globally and has impacted all features of healthcare worldwide, including the delivery of care to patients with fertility-related issues. Recent studies state that infertile persons frequently expressed added levels of anxiety and depression compared to the general population during the COVID-19 era [37–41].

Sub-objectives were the determination of the factors that influence the quality of life of infertile couples; and the social, family, and demographic relationships that are affected by infertility and have an impact on the quality of life.

## 2. Materials and Methods

The psychometric tools used in this research to collect the data were self-administered questionnaires: The Demographic Information and Medical History Questionnaire—a COMPI Questionnaire (Copenhagen Multi-Centre Psychosocial Infertility Research Programme); and the FertiQoL International Questionnaire for measuring the quality of life. The collection of the questionnaires took place during the period of January–November 2022.

### 2.1. Sample

The study sample consists of 101 infertile couples (101 women and 101 men) who were undergoing assisted reproduction techniques at the Athens Naval Hospital-Assisted Reproduction Unit (ANH-ARU) during the first eleven months of 2022 (1 January 2022–30 November 2022).

Men and women who live in the wider area of Attica and in various cities in Greece come to the ANH-ARU.

### 2.2. Data Collection Process

Self-administered questionnaires were delivered in a paper-and-pen format to eligible participants. Participation in the survey was voluntary. A short paragraph was included at the beginning of the questionnaires to inform participants of the study's objectives and their responses' confidentiality. All participants gave informed consent. Data were collected anonymously, and participants had the right to access their answers and withdraw from the research whenever they wished to. The research protocol was approved by the Athens Naval Hospital Research Ethics Committee (ANH-RPA-325/30-12-2021). Contact details of the researcher, such as telephone and email, were also provided.

Due to the fact that filling out the questionnaires was time-consuming, it was deemed appropriate for the health recipients to have the option of either filling in the questionnaires immediately or filling in the questionnaires outside of the ANH–ARU. The completed questionnaires were returned to the ANH–ARU during the next appointment in person.

It was considered appropriate for couples to fill in the medical history at the ANH–ARU to avoid incorrect answers. Furthermore, the researcher gave a series of instructions regarding the filling in of the questionnaires, such as urging the couples to answer with

spontaneity and honesty and to not fill in the questionnaires in collaboration with their respective partners.

### 2.3. Ethics and Ethics Protection of Personal Data

A request for approval was made to the scientific council of the Athens Naval Hospital in order to approve the study protocol and methodology and the compliance with the General Data Protection Regulation (GDPR). The questionnaires were accompanied by an information brochure in which it was stated in detail that the choice to participate in the research was entirely optional and would have no effect on their fertility treatment.

All personal data and the answers given remained accessible only to the main researcher. Responses were strictly confidential and were destroyed after the survey was completed. Strict confidentiality, and of course, the anonymity of the participants, were maintained. Furthermore, they had the right to stop their participation in the research at any time they wanted and were informed about the aims and purposes of the research.

### 2.4. Statistical Analysis

Quantitative variables are expressed as mean values (standard deviation) and as median (interquartile range); qualitative variables are expressed as absolute and relative frequencies. Pearson's (r) and Spearman's (rho) correlations coefficients were used to exploring the association of two continuous variables. Multiple linear regression analysis was used with dependence on the FERTI-QoL subscales. The regression equation included terms for the participant's demographics and information from his medical history. Adjusted regression coefficients (β) with standard errors (SE) were computed from the results of the linear regression analyses. All reported p values are two-tailed. Statistical significance was set to $p < 0.05$, and analyses were conducted using SPSS statistical software (version 22.0).

### 3. Results

The sample consisted of 202 spouses, i.e., 101 couples, with a mean age of 39.5 years (SD = 4.9 years). Their characteristics are presented in Table 1. Among them, 7.9% already had a child. Additionally, 28.7% were high school graduates, 51.5% had 1001 to 1300 euros as their monthly income, and 2.5% suffered from a chronic disease. Moreover, 45.5% of the participants had been pregnant in the past, of which 16.0% had a baby, 54.3% had a miscarriage, and 77.2% had been under treatment for infertility. They had been trying for a baby for a median time of three years (IQR: 2–4 years). The mean state score was 39.3 (SD = 10.5), the mean trait was 36.6 (SD = 9.7), and the median depression score was 5 (IQR: 0–8).

The mean overall FERTI-QoL score was 70.2% (SD = 13.8); see Table 2. Greater anxiety and depression were significantly associated with worse quality of life.

When multiple regression analysis was conducted, the lowest score was found for the "emotional" subscale, indicating worse emotional quality of life for women, participants with higher levels of education, those who suffered from s chronic somatic disease, those who had experienced a miscarriage, and those who had been under treatment for infertility (Table 3). Additionally, greater state and depression scores were significantly associated with worse quality in the "emotional" subscale. On the contrary, a greater score in the "emotional" subscale was significantly associated with more years of romantic relationship with one's spouse. Furthermore, significantly lower scores for the "Mind-body" subscale, indicating worse quality, were had by women, participants with higher levels of education, and those who had experienced a miscarriage. Additionally, greater state and depression scores were significantly associated with worse quality in the "Mind-body" subscale. Significantly lower scores in the "Relation" subscale, indicating worse quality in that specific sector, were had by women, participants who had been pregnant, and those with greater depression symptoms. In addition, significantly lower scores in the "Social" subscale, indicating worse quality in that specific sector, were had by women, participants

with higher levels of education, those who were trying more years to have a baby, and those with greater depression symptoms.

**Table 1.** Sample characteristics.

| | N (%) |
|---|---|
| Age, mean (SD) | 39.5 (4.9) |
| Gender | |
|     Men | 101 (50.0) |
|     Women | 101 (50.0) |
| Children | 16 (7.9) |
| Educational level | |
|     Primary school | 0 (0.0) |
|     Middle school | 3 (1.5) |
|     High school | 58 (28.7) |
|     2-year college | 37 (18.3) |
|     Technical university | 46 (22.8) |
|     University | 25 (12.4) |
|     MSc/PhD | 33 (16.3) |
| Chronic somatic disease | 5 (2.5) |
| Monthly income | |
|     Up to 700.00€ | 45 (22.3) |
|     701.00–1000.00€ | 21 (10.4) |
|     1001.00–1300.00€ | 104 (51.5) |
|     1301.00–1500.00€ | 23 (11.4) |
|     1501.00–2000.00€ | 1 (0.5) |
|     >2000.00€ | 8 (4.0) |
| Years with a spouse, mean (SD) | 7.1 (2.9) |
| Ever been (yourself or your spouse) pregnant | 46 (45.5) |
| Ever had (yourself or your spouse) a baby | 8 (16) |
| Ever had (yourself or your spouse) a miscarriage | 25 (54.3) |
| Years of trying to have a baby, median (IQR) | 3 (2–4) |
| Ever been (you or your spouse) under treatment for infertility | 78 (77.2) |
| State, mean (SD) | 39.3 (10.5) |
| Trait, mean (SD) | 36.6 (9.7) |
| Depression score, median (SD) | 5 (0–8) |

**Table 2.** Descriptive statistics of FERTI-QoL scales and their correlations with anxiety and depression scales.

| | | Correlation Coefficients | | |
|---|---|---|---|---|
| | **Mean (SD)** | **State [1]** | **Trait [1]** | **Depression [2]** |
| Emotional | 62.3 (20.5) | −0.39 *** | −0.33 *** | −0.34 *** |
| Mind-body | 71.7 (21.4) | −0.50 *** | −0.43 *** | −0.37 ** |
| Relation | 77.5 (15.2) | −0.38 *** | −0.38 *** | −0.22 ** |
| Social | 68.5 (18.4) | −0.07 | −0.14 * | −0.32 *** |
| Environment | 73.1 (13.9) | −0.36 *** | −0.21 ** | 0.05 |
| Tolerability | 68.1 (25.0) | −0.44 *** | −0.32 ** | −0.35 *** |
| Core | 70.0 (14.8) | −0.43 *** | −0.41 *** | −0.36 *** |
| Treatment | 71.1 (15.6) | −0.48 *** | −0.32 *** | −0.24 ** |
| Total FERTIQOL score | 70.3 (13.8) | −0.49 *** | −0.42 *** | −0.3 6*** |

[1] Pearson's correlation coefficient; [2] Spearman's correlation coefficient. * $p < 0.05$; ** $p < 0.01$; *** $p < 0.001$.

Lower scores for the "Environment" subscale, indicating worse quality in that specific sector, were had by participants with higher levels of education or greater monthly income and those with greater state scores (Table 4). Moreover, significantly lower scores in the "Tolerability" and in the "Core" subscale, indicating worse quality in those specific sectors, were had by women, participants with higher levels of education, those with greater depressive symptoms, and those with greater state scores. Similarly, significantly

lower scores in the "Treatment" subscale, indicating worse quality in that specific sector, were had by women, participants with higher levels of education, and those with greater state scores.

**Table 3.** Multiple linear regression results for "Emotional", "Mind-body", "Relation", and "Social" subscales.

| | Emotional | | Mind-Body | | Relation | | Social | |
|---|---|---|---|---|---|---|---|---|
| | β (SE) + | *p* | β (SE) + | *p* | β (SE) + | *p* | β (SE) + | *p* |
| Age | 0.35 (0.24) | 0.153 | 0.23 (0.23) | 0.323 | −0.05 (0.21) | 0.797 | 0.41 (0.25) | 0.097 |
| Gender (women vs. men) | −14.73 (2.61) | <0.001 | −13.59 (2.46) | <0.001 | −4.96 (2.3) | 0.032 | −8.57 (2.64) | 0.001 |
| Children (yes vs. no) | 6.2 (11.29) | 0.584 | 16.47 (10.64) | 0.123 | 7.56 (9.94) | 0.448 | 3.86 (11.41) | 0.736 |
| Educational level | −2.42 (0.86) | 0.006 | −2.92 (0.81) | <0.001 | 0.24 (0.76) | 0.751 | −2.16 (0.87) | 0.014 |
| Chronic somatic disease (yes vs. no) | −17.20 (7.18) | 0.018 | −7.02 (6.77) | 0.301 | −5.94 (6.32) | 0.348 | −5.81 (7.26) | 0.425 |
| Monthly income | 0.89 (1.26) | 0.480 | −0.53 (1.19) | 0.654 | 0.23 (1.11) | 0.834 | −2.24 (1.27) | 0.081 |
| Years with spouse | 1.08 (0.53) | 0.043 | −0.16 (0.5) | 0.745 | 0.86 (0.47) | 0.068 | 1 (0.53) | 0.063 |
| Ever been (yourself or your spouse) pregnant (yes vs. no) | 0.57 (4.23) | 0.893 | 4.31 (3.98) | 0.105 | −10.51 (3.72) | 0.005 | −0.89 (4.27) | 0.835 |
| Ever had (yourself or spouse) a baby (yes vs. no) | −9.34 (11.95) | 0.436 | −10.1 (11.26) | 0.371 | 6.33 (10.52) | 0.548 | 5.13 (12.08) | 0.671 |
| Ever had (yourself or spouse) a miscarriage (yes vs. no) | −8.54 (4.04) | 0.036 | −13.81 (3.81) | <0.001 | 8.32 (4.56) | 0.070 | −2.9 (4.08) | 0.479 |
| Years of trying to have a baby | 0.02 (0.85) | 0.984 | −0.49 (0.8) | 0.540 | 0.2 (0.75) | 0.787 | −2.15 (0.86) | 0.013 |
| Ever been (you or your spouse) under treatment for infertility (yes vs. no) | −9.31 (3.64) | 0.011 | −4.1 (3.43) | 0.233 | 2.11 (3.2) | 0.511 | 3.81 (3.67) | 0.302 |
| State | −0.56 (0.20) | 0.006 | −0.94 (0.19) | <0.001 | −0.24 (0.18) | 0.182 | 0.17 (0.2) | 0.406 |
| Trait | 0.19 (0.23) | 0.401 | 0.2 (0.21) | 0.341 | −0.12 (0.2) | 0.551 | 0.14 (0.23) | 0.555 |
| Depression score | −0.58 (0.23) | 0.013 | −0.64 (0.22) | 0.004 | −0.62 (0.2) | 0.003 | −1.39 (0.24) | <0.001 |

+ regression coefficient (standard error).

**Table 4.** Multiple linear regression results for "Environment", "Tolerability", "Core", and "Treatment" subscales.

| | Environment | | Tolerability | | Core | | Treatment | |
|---|---|---|---|---|---|---|---|---|
| | β (SE) + | *p* | β (SE) + | *p* | β (SE) + | *p* | β (SE) + | *p* |
| Age | −0.01 (0.19) | 0.940 | −0.01 (0.27) | 0.972 | 0.23 (0.17) | 0.169 | −0.01 (0.19) | 0.947 |
| Gender (women vs. men) | −1.4 (2.05) | 0.496 | −19.22 (2.91) | <0.001 | −10.46 (1.81) | <0.001 | −8.53 (2.01) | <0.001 |
| Children (yes vs. no) | −9.71 (8.87) | 0.275 | −5.01 (12.61) | 0.692 | 8.52 (7.82) | 0.277 | −7.83 (8.72) | 0.370 |
| Educational level | −1.91 (0.68) | 0.005 | −5.14 (0.96) | <0.001 | −1.82 (0.6) | 0.003 | −3.20 (0.67) | <0.001 |
| Chronic somatic disease (yes vs. no) | 8.95 (5.65) | 0.115 | −1.28 (8.02) | 0.873 | −8.99 (4.98) | 0.072 | 4.86 (5.55) | 0.382 |
| Monthly income | −2.2 (0.99) | 0.028 | 0.38 (1.41) | 0.789 | −0.41 (0.87) | 0.638 | −1.17 (0.97) | 0.231 |
| Years with spouse | 0.56 (0.42) | 0.180 | −0.46 (0.59) | 0.441 | 0.69 (0.37) | 0.060 | 0.15 (0.41) | 0.708 |
| Ever been (yourself or your spouse) pregnant (yes vs. no) | −4.24 (3.32) | 0.203 | 7.28 (4.72) | 0.125 | 0.12 (2.93) | 0.967 | 0.37 (3.26) | 0.911 |
| Ever had (yourself or spouse) a baby (yes vs. no) | 9.50 (9.39) | 0.313 | 17.3 (13.35) | 0.197 | −1.99 (8.28) | 0.810 | 12.62 (9.23) | 0.173 |
| Ever had (yourself or spouse) a miscarriage (yes vs. no) | 4.45 (3.18) | 0.163 | −2.19 (4.52) | 0.629 | −4.23 (2.80) | 0.132 | 1.79 (3.12) | 0.566 |
| Years of trying to have a baby | −0.41 (0.67) | 0.544 | 0.04 (0.95) | 0.969 | −0.61 (0.59) | 0.305 | −0.23 (0.66) | 0.728 |
| Ever been (you or your spouse) under treatment for infertility (yes vs. no) | 3.08 (2.86) | 0.283 | 0.20 (4.06) | 0.961 | −1.87 (2.52) | 0.458 | 1.93 (2.81) | 0.494 |
| State | −0.40 (0.16) | 0.013 | −1.07 (0.23) | <0.001 | −0.39 (0.14) | 0.006 | −0.67 (0.16) | <0.001 |
| Trait | 0.03 (0.18) | 0.882 | 0.46 (0.25) | 0.070 | 0.10 (0.16) | 0.514 | 0.20 (0.18) | 0.253 |
| Depression score | 0.26 (0.18) | 0.150 | −0.52 (0.26) | 0.048 | −0.81 (0.16) | <0.001 | −0.05 (0.18) | 0.788 |

+ regression coefficient (standard error).

Overall, quality of life, according to FERTI-QoL, was significantly worse in women, participants with higher levels of education, those with greater depressive symptoms, and those with greater state scores (Table 5).

**Table 5.** Multiple linear regression results for overall FERTI-QoL score.

| | Total FERTIQOL Score | |
|---|---|---|
| | β (SE) + | *p* |
| Age | 0.16 (0.15) | 0.299 |
| Gender (women vs. men) | −9.89 (1.65) | <0.001 |
| Children (yes vs. no) | 3.71 (7.15) | 0.604 |
| Educational level | −2.22 (0.55) | <0.001 |
| Chronic somatic disease (yes vs. no) | −4.92 (4.55) | 0.281 |
| Monthly income | −0.64 (0.80) | 0.428 |
| Years with spouse | 0.53 (0.34) | 0.113 |
| Ever been (yourself or your spouse) pregnant (yes vs. no) | 0.19 (2.68) | 0.943 |
| Ever had (yourself or spouse) a baby (yes vs. no) | 2.3 (7.57) | 0.761 |
| Ever had (yourself or spouse) a miscarriage (yes vs. no) | −2.46 (2.56) | 0.338 |
| Years of trying to have a baby | −0.50 (0.54) | 0.360 |
| Ever been (you or your spouse) under treatment for infertility (yes vs. no) | −0.76 (2.30) | 0.743 |
| State | −0.47 (0.13) | <0.001 |
| Trait | 0.13 (0.14) | 0.361 |
| Depression score | −0.58 (0.15) | <0.001 |

+ regression coefficient (standard error).

## 4. Discussion

According to the research results, as they were studied and analyzed, the research hypotheses can be verified. The study showed that stress and infertility are related to the quality of life and that the higher the levels of stress and anxiety, the lower the level of quality of life. Women experience stress at higher levels, and this can be seen in the fact that they show intense symptoms of anxiety and depression compared to men, negatively affecting their quality of life.

Conflicts in the couple increase but are easier to deal with when the couple has been together for several years.

It is worth noting that in this research, the higher the educational level, the worse the couple's quality of life. Higher professional expectations impact one's reproductive behavior due to the intense goals beyond those related to the family. A higher level of education impacts their attitudes and preferences as individuals. Perhaps the most important of these changes is related to the fact that a higher level of education is associated with greater and more competitive participation of individuals in the labor market.

Regarding sociability and the appearance of the problem in the rest of the social environment, the results show that social interactions are affected by fertility problems, such as social inclusion, expectations, stigma, and support. Similar results in recent studies [42–44] state that women believe that they should discuss more with their own family members and not so much with their husbands, as the highest levels of support and understanding of their childlessness come initially from their own blood relatives.

It is important to mention that women with a high level of education and those who had experienced a miscarriage showed the impacts of fertility problems on physical health, cognition, and behavior. Bearing a child is often their life goal, and women focus all their efforts on this, as they consider themselves biologically unsuccessful without children.

Greece is facing a dynamic, protracted, and complex demographic problem. Of particular concern is the long-term course of very low fertility, which is intertwined with increasing rates of infertility. According to data from the European Statistical Service (Eurostat), the above is also confirmed by the fertility index of the countries of the European Union (March 2019), which ranks Greece in last place [45].

Through the results of this research, an attempt was made to better understand the psychology of infertility and how infertility affects the couple's quality of life.

The evaluation of the conclusions can be useful to health professionals working in hospitals, obstetric clinics, and specialized artificial insemination centers. The results are particularly useful regarding the approach for psychological support to achieve the intended upgrade in the quality of life of infertile couples.

The present results could be used by midwives and health professionals involved in counseling infertile couples to help them externalize their feelings and better understand the experience of impaired fecundity. Based on the latest data mentioned, it is necessary to establish organizations and social actions with a supporting and advisory role for couples facing involuntary childlessness. The findings of the study highlight the need to implement interventions both for the early identification of infertile couples who present low levels of quality of life and to reduce the effects of infertility stress on these people.

The working environment of the armed forces presents some peculiarities compared to other work environments (public or private sector). Combat stress or operational stress is defined as the internal process that prepares the combatant's psychosomatic response to the changing and life-threatening conditions in which he performs his mission. It can undermine his judgment, performance, and effectiveness [46].

Our study sample came from the medically assisted reproduction unit of Athens Naval Hospital. In this particular unit, those serving in the Hellenic Armed Forces and Security Forces (Army, Navy, Air Force, Coast Guard, Hellenic Police, Fire Service), and their family members, are entitled to free healthcare. Officers serving in the Armed Forces and Security Forces face the demands of operational life and are exposed to a stressful military work environment [47].

In 2018, the Service Women's Action Network (SWAN) conducted an online survey focused on reproductive health services in the military. Of the 799 total surveys of active-duty service women who answered questions about infertility, 37% said that they had trouble getting pregnant when actively trying to do so [48]. The results of this survey caused concern about military leadership, as the findings suggested a much higher prevalence of female infertility among service women compared to the Centers for Disease Control and Prevention's (CDC's) national prevalence estimate. According to the CDC's 2011–2015 National Survey of Family Growth, the prevalence of infertility among married women 15–44 years old was 6.7%; 12.1% of women aged 15–44 years reported impaired fecundity [49]. The CDC defined infertility as a self-report of at least 1 year of failed attempts by married/cohabiting partners at getting pregnant when neither the respondent nor her current husband/cohabiting partner was surgically sterile and when the couple had been sexually active each month without contraception [49]. Impaired fecundity was defined as self-reported problems getting pregnant and carrying a baby to term regardless of marital/cohabiting status [49]. It has been suggested that service women may be at increased risk for infertility because of exposures to environmental toxins and traumas and/or stressors experienced during deployments [48,50,51]. In addition, relatively higher levels of tobacco use, alcohol use, and pelvic inflammatory disease (PID) also may put service women at greater risk for infertility than the national female population [52–54].

Regarding the limitations of this particular study, the core one is the relatively small sample (200 people) used to conduct the research, from which the results and conclusions were derived, due to the suspension of medically assisted reproduction units during the period (March–April 2022) of the emergence and spread of the SARS-CoV-2 virus, following the recommendations of E.S.H.R.E, A.S.R.M., and W.H.O.

Additionally, the sample came only from the medically assisted reproduction unit of Athens Naval Hospital, so the sample could not be representative, and the results cannot be generalized to the entire Greek population. However, it appears that the infertile sample consisted of women and men with varied demographic and medical characteristics.

The sample of the current research involved only participants who had decided to seek assisted reproductive treatment. Thus, the investigation of the relationship between

quality of life and infertility stress was examined using only a sample of infertile couples who were about to undergo or had previously undergone fertility treatment.

During the completion of the questionnaires, no specialized laboratory tests were carried out to achieve an accurate measurement of stress levels. Participants' reports and statements of stress levels as indicators of quality of life are likely to have been subjective.

What is less clear is whether stress could be a cause of infertility. Its potential impact is difficult to explore because stress is subjective and difficult to measure, and when starting infertility treatments, most couples are optimistic about their outcomes. Thus, it is not easy to conduct long-term, reliable studies investigating fertility before and after the onset of stress. However, the existing, albeit small, studies suggest that stress is probably an aggravating factor that negatively affects the quality of life.

Finally, an important factor while conducting this research (January–November 2022) is the continuity of SARS-CoV-2's spread, along with the efforts to control the infection rates and the progress of the pandemic. Research till now [37–39,41] on experiences of COVID-19 in the general population and infertile people shows more anxiety and depression among respondents than historical norms.

Despite these limitations, the study had several strengths. The sample size was sufficient to achieve all research objectives, study the effects of stress and infertility on couples' quality of life, and test all research hypotheses amidst the COVID-19 pandemic.

An advantage of the present study is the fact that although many studies have been conducted on the analysis of quality of life in infertile individuals, most have focused on one of the two spouses, especially women. This study also investigated the role of infertility and stress in couples' quality of life as a dyad.

The questionnaires and scales used are valid, international, weighted to Greek data, have satisfactory psychometric properties, and have been reused in similar studies.

Sophisticated statistical tools were used to make the analyses and correlations.

## 5. Conclusions

It is not clear whether an elevated level of distress arises in all infertile couples. The level of stress and changes in QoL may be related to non-medical conditions, and factors predicting QoL may vary in different infertile populations and genders. Therefore, the identification of aspects related to better or worse health-related QoL is crucial for suggesting and testing scientific interventions for infertile populations. Based on the conclusions drawn, proposals can be formulated for the benefits of a short-term stress management program that will contribute to improving the quality of life of those facing infertility.

Therefore, suggestions for clinical practice generally concern the development of methods and interventions to reduce psychological stress in infertile individuals undergoing fertility treatment, considering the particularities that require changing their behavior beyond the usual routine.

The present results could also be utilized to provide information, support, and treatment of the stress experienced by female army officers who seek treatment options to get pregnant.

**Author Contributions:** Conceptualization, P.D. and A.L.; data curation, P.D.; formal analysis, P.D., K.G. (Kleanthi Gourounti) and A.S.; Funding acquisition, P.D.; Investigation, P.D. and K.G. (Konstantina Gaitanou).; methodology, P.D. and A.S.; project administration, P.D., K.G. (Konstantina Gaitanou) and N.P.; supervision, A.S.; validation, A.S.; writing—original draft, P.D. and A.S.; writing—review and editing, A.S. All authors have read and agreed to the published version of the manuscript.

**Funding:** The APC was funded by the Special Account for Research Grants, University of West Attica, Athens, Greece.

**Institutional Review Board Statement:** The study was conducted in accordance with the Declaration of Helsinki and approved by Athens Naval Hospital Research Ethics Committee (ANH-RPA-325/30-12-2021).

**Informed Consent Statement:** Informed consent was obtained from all subjects involved in the study.

**Data Availability Statement:** The datasets generated during and/or analyzed during the current study are not publicly available; however, they are available from the corresponding author upon reasonable request.

**Conflicts of Interest:** The authors declare no conflict of interest.

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
