# Peer review of "Quality of Life among Couples with a Fertility Related Diagnosis"

_clinpract, doi:10.3390/clinpract13010023_

Round 1

Reviewer 1 Report

I found this article to be interesting and novel in terms of considering QoL in couples. The sample size was sufficient.  However, certain sections were written in a confusing way and need to be reviewed. 

"Therefore, the question that arises is whether psychological stress and its consequences adversely affect fertility. However, all researchers come to a common conclusion".  The authors make this statement but do not state what the conclusion is. 

Throughout the question of whether fertility is affected by stress is raised a few times but I feel that this is not done in the right context. This paper does not address this issue at all so this needs to be stated at the beginning.  Impact of stress on fertility is not the hypothesis. 

"There are many ways and tools to measure the psychological burden and quality of life associated with infertility and the in vitro fertilization process as well as several treatment options. Interventions to reduce and alleviate the various clinical symptoms, regardless of whether or not they contribute to an increase in pregnancy rates, are extremely beneficial to the emotional balance and harmonious relationship of the couple [10]."  This feels like a series of unrelated sentences. What are the many ways and tools - these need to be referenced. There is only one reference to the second statement.  

"Although official data indicate that 10%-15% of the world's general population of reproductive age faces infertility [11], the emotions created in the couple when faced with infertility are particularly intense and can lead to an unexpected emotional crisis".  The first part of the sentence does not relate to the second part of the statement. 

"Some studies have shown that infertility leads to intense stress and a decrease in love  between infertile couples." These studies need to be referenced. 

Line 219 - Children had 7.9% of the 219 sample.  Does this mean 7.9% of the sample had a child already?

Line 282 - Therefore, if the acquisition of a child is the goal, it will hardly be anything, an obstacle to the completion of this goal.  What does this sentence mean?

Conclusion - "Based on the conclusions drawn, proposals can be formulated for the benefits of a short-term stress management program that will contribute as a valuable tool to operational readiness and the upgrading of the executives' quality of life".  This statement feels cut and pasted. This is the first mention of executives - what does this mean?

Conclusion - The present results could be utilized in order to provide information, support, and  treatment of the daily and work stress of the armed force officers.  I don't think this is what the results say. the results say it is the women who are most affected not the armed force officers. 

The conclusion is very vague. 

Author Response

Dear Reviewer

Thank you for giving us the opportunity to submit a revised draft of our manuscript titled

“Quality of Life Among Couples with fertility related diagnosis” to “Clinics and Practice”. We highly appreciate the time and effort that you have dedicated to providing your valuable feedback on our manuscript. We are grateful to the reviewers for their insightful comments. We have been able to incorporate changes in the manuscript to reflect most of your suggestions and they are marked in yellow.

In addition to the above comments, all spelling and grammatical errors pointed out by the reviewers have been corrected.

We look forward to hearing from you in due time regarding our submission and responding to any further suggestions and comments you may have.

Sincerely,

A. Sarantaki

REVIEWER 1

Comment 1: "Therefore, the question that arises is whether psychological stress and its consequences adversely affect fertility. However, all researchers come to a common conclusion".  The authors make this statement but do not state what the conclusion is.

Response: Thank you for your comment. We have referenced the researchers of the studies and stated what the conclusion is.

"Therefore, the question that arises is whether psychological stress and its consequences adversely affect fertility. The research evidence, regarding the statistically significant relationship between stress and infertility, is contradictory [10]. However, all researchers come to a common conclusion that regardless of the positive or negative impact of stress levels on infertility treatment, one of the primary goals of health professionals dealing with the issue should be to provide valid information and advice as well as providing psychological support. Infertility and its treatment are a couple's problem and not an individual one and should be treated as such. Not all individuals are equally susceptible to having their mental health disrupted by chronic infertility. However, if it is perceived that their mental balance and interpersonal relationship are disturbed, they should consult specialized health professionals [7]."

Comment 2: Throughout the question of whether fertility is affected by stress is raised a few times but I feel that this is not done in the right context. This paper does not address this issue at all so this needs to be stated at the beginning.  Impact of stress on fertility is not the hypothesis.

Response : Thank you for your comment. We have added the following paragraph to the introduction

"In addition, there is a new definition in the literature for the fertility quality of life (FertiQoL), specifically evaluating the impact of fertility problems on various life dimensions [7]. Infertile women frequently report severe stress, poorer marital adjustment and QoL than women who conceive naturally [8]. Moreover, men may experience less intercourse satisfaction, perhaps because of the psychological pressure of trying to conceive or the forced timing of intercourse around the woman’s ovulatory cycle [8]."

Comment 3: "There are many ways and tools to measure the psychological burden and quality of life associated with infertility and the in vitro fertilization process as well as several treatment options. Interventions to reduce and alleviate the various clinical symptoms, regardless of whether or not they contribute to an increase in pregnancy rates, are extremely beneficial to the emotional balance and harmonious relationship of the couple [10]."  This feels like a series of unrelated sentences. What are the many ways and tools - these need to be referenced. There is only one reference to the second statement. 

Response : Thank you for your remark. We have referenced the second statement and the ways and tools to measure the psychological burden and quality of life.

"There are many ways and research tools to measure the psychological burden and quality of life associated with infertility and the in vitro fertilization process as well as several treatment options [10]. Tools to measure health and quality of life can be general or focus on specific diseases. They combine the subjective dimension of the individual's personal experience and the objective assessment of external factors that can affect the quality of life of the individual. The subjective indicators are related to a person’s satisfaction with life, the feeling of adequacy and satisfaction from its functionality in several areas of life, satisfaction from services provided, and the possibility to participate in various activities (recreational, educational). Health, natural environment, space and living conditions, time availability, social activity, financial ability to cover basic needs, its security environment, are the objective indicators that are valued [11].

Interventions to reduce and alleviate the various clinical symptoms, regardless of whether or not they contribute to an increase in pregnancy rates, are extremely beneficial to the emotional balance and harmonious relationship of the couple [12]."

Comment 4: "Although official data indicate that 10%-15% of the world's general population of reproductive age faces infertility [11], the emotions created in the couple when faced with infertility are particularly intense and can lead to an unexpected emotional crisis".  The first part of the sentence does not relate to the second part of the statement.

Response:  Thank you for your comment. We have, accordingly, modified the sentence so that it makes more sense.

"Data indicate that 10%-15% of the global population of reproductive age experience infertility [13], yet the emotional distress of this diagnosis can be especially intense and cause an unexpected crisis [14]."

Comment 5: “Some studies have shown that infertility leads to intense stress and a decrease in love  between infertile couples." These studies need to be referenced.

Response : Thank you for pointing this out. We have referenced the study.

Some studies have shown that infertility leads to intense stress and a decrease in affection between infertile couples [15].

Comment 6: Line 219 - Children had 7.9% of the 219 sample.  Does this mean 7.9% of the sample had a child already?

Response: We have made the change. We put the alteration into effect. The revised phrase is as follows.

“7.9% of the sample had already a child”

Comment 7: Line 282 - Therefore, if the acquisition of a child is the goal, it will hardly be anything, an obstacle to the completion of this goal.  What does this sentence mean?

Response: This observation is correct. We implemented the alteration. The revised statement is as follows.

“However, bearing a child is often their life goal, and focus all their efforts on this as they consider themselves biologically unsuccessful without children.”

Comment 8: "Based on the conclusions drawn, proposals can be formulated for the benefits of a short-term stress management program that will contribute as a valuable tool to operational readiness and the upgrading of the executives' quality of life".  This statement feels cut and pasted. This is the first mention of executives - what does this mean?

Response: We agree and have updated the statement as follows: “It is not clear whether an elevated level of distress arises in all infertile couples. The level of stress and changes in QoL may be related to non-medical conditions and factors predicting QoL may vary in different infertile populations and genders. Therefore, the identification of aspects related to better or worse health-related QoL is crucial for suggesting and testing scientific interventions for infertile populations. Based on the conclusions drawn, proposals can be formulated for the benefits of a short-term stress management program that will contribute to improving the quality of life of those facing infertility.”

Comment 9: “The present results could be utilized in order to provide information, support, and treatment of the daily and work stress of the armed force officers.  I don't think this is what the results say. the results say it is the women who are most affected not the armed force officers.”

Response: We are appreciative of your observation, and we concur with your comment. We have changed to: “The present results could also be utilized to provide information, support, and treatment of the stress experienced by female army officers who seek treatment options to get pregnant.”

We have also revised the conclusion.

Reviewer 2 Report

 This is a valuable study of QoL among infertile couples. The authors acknowledge that the self-reported answers in the survey are likely to be subjective. And this is my primary concern with the study.

 For example:  No attempts have been made in this study to measure the deployment-related base-line stress among those spouses who were deployed, had recently returned from deployment, or were about to be deployed around the time of the study. The baseline stress is likely to affect the outcome of self-reported responses to questions. Assessment of prior or existing stress levels, especially in the military setting, would have made this study more meaningful. It would be helpful if the authors comment in th Discussion section on why this was not done.

Author Response

Dear Reviewer

Thank you for giving us the opportunity to submit a revised draft of our manuscript titled

“Quality of Life Among Couples with fertility related diagnosis” to “Clinics and Practice”. We deeply appreciate the time and effort that you have dedicated to providing your valuable feedback on our manuscript. We are grateful to the reviewers for their insightful comments. We have been able to incorporate changes in the manuscript to reflect most of your suggestions and they are marked in yellow.

In addition to the above comments, all spelling and grammatical errors pointed out by the reviewers have been corrected.

We look forward to hearing from you in due time regarding our submission and responding to any further suggestions and comments you may have.

Sincerely,

A. Sarantaki

REVIEWER 2

Comment: "This is a valuable study of QoL among infertile couples. The authors acknowledge that the self-reported answers in the survey are likely to be subjective. And this is my primary concern with the study.

 For example:  No attempts have been made in this study to measure the deployment-related base-line stress among those spouses who were deployed, had recently returned from deployment, or were about to be deployed around the time of the study. The baseline stress is likely to affect the outcome of self-reported responses to questions. Assessment of prior or existing stress levels, especially in the military setting, would have made this study more meaningful. It would be helpful if the authors comment in the Discussion section on why this was not done."

Response: Thank you for pointing this out. We agree with this comment. Therefore, we address the issue of baseline stress in the Discussion section, especially in the military setting, by noting some references such as:

 In 2018, the Service Women’s Action Network (SWAN) conducted an online survey focused on reproductive health services in the military. Of the 799 total survey of the active duty service women who answered questions about infertility, 37% said that they had trouble getting pregnant when actively trying to do so. (Service Women Action Network (SWAN). Access to reproductive health care: the experience of military women. https://www.servicewomen.org/wp-content/uploads/2018/12/2018Reproreport SWAN-2.pdf.Accessed 14 April 2019). The results of this survey caused concern about military leadership, as the findings suggested a much higher prevalence of female infertility among service women compared to the Centers for Disease Control and Prevention’s (CDC’s) national prevalence estimate. According to the CDC’s 2011-2015 National Survey of Family Growth, the prevalence of infertility among married women 15-44 years old was 6.7%; 12.1% of women aged 15-44 years reported impaired fecundity. (Centers for Disease Control and Prevention. National Center for Health Care Statistics. Key Statistics from the National Survey of Family Growth-I Listing. https://www.cdc.gov/nchs/nsfg/key statistics/i.htm. Accessed 14April 2019). The CDC defined infertility as a self-report of at least 1 year of failed attempts for married/cohabitating partners at getting pregnant when neither the respondent nor her current husband/cohabiting  partner was surgically sterile and when the couple had been sexually active each month without contraception. (Centers for Disease Control and Prevention. National Center for Health Care Statistics. Key Statistics from the National Survey of Family Growth-I Listing. https://www.cdc.gov/nchs/nsfg/key statistics/i.htm. Accessed 14April 2019). Impaired fecundity was defined as self-reported problems getting pregnant and carrying a baby to term regardless of marital/cohabitating status (Centers for Disease Control and Prevention. National Center for Health Care Statistics. Key Statistics from the National Survey of Family Growth-I Listing. https://www.cdc.gov/nchs/nsfg/key statistics/i.htm. Accessed 14April 2019). It has been suggested that service women may be at increased risk for infertility because of exposures to environmental toxins as well as traumas and/or stressors experienced during deployments. (Service Women Action Network (SWAN). Access to reproductive health care: the experience of military women. https://www.servicewomen.org/wp-content/uploads/2018/12/2018Reproreport SWAN-2.pdf.Accessed 14 April 2019). (Rooney KL, Domar AD. The relationship between stress and infertility. Dialogues Clin Neurosci. 2018 Mar;20(1):41-47) (Armed Forces Health Surveillance Center. Health of women after wartime deployments:correlates of risk for selected medical conditions among females after initial and repeat deployments to Afghanistan and Iraq, active component, U.S. Armed Forces. MSMR. 2012;19(7):2-10). In addition, relatively higher levels of tobacco use, alcohol use, and pelvic inflammatory disease (PID) also may put service women at greater risk for infertility than the national female population. (Van Heertum K, Rossi B. Alcohol and Fertility: how much is too much? Fertil Res Pract. 2017;3:10) (McKee DL, Hu Z, Stahlman S. Incidence and sequelae of acute pelvic inflammatory disease among active component females, U.S Armed Forces, 1996-2016. MSMR. 2018;25(10):2-8) (Stahlman S, Oetting AA. Mental health disorders and mental health problems, active component, U.S. Armed Forces, 2007-2016. MSRM. 2018;25(3):2-11).

We also suggest that future studies could benefit from assessing prior or existing stress levels in order to gain a better understanding of the impact of infertility on QoL.